# Maternal Effects of Habitats Induce Stronger Salt Tolerance in Early-Stage Offspring of *Glycyrrhiza uralensis* from Salinized Habitats Compared with Those from Non-Salinized Habitats

**DOI:** 10.3390/biology13010052

**Published:** 2024-01-19

**Authors:** Junjun Gu, Shaoxuan Yao, Miao Ma

**Affiliations:** Key Laboratory of Xinjiang Plant Medicinal Resources Utilization, Ministry of Education, College of Life Sciences, Shihezi University, Shihezi 832003, China; gujunjun0120@163.com (J.G.); yaoshaoxuan199811@163.com (S.Y.)

**Keywords:** *G. uralensis*, salt stress, seeds, maternal effect, antioxidant enzymes, reactive oxygen species

## Abstract

**Simple Summary:**

Differences in the responses of early-stage offspring seeds from different habitats to salt are unclear. We evaluated the salt tolerance of *Glycyrrhiza uralensis* Fisch (licorice) germplasms by comparing differences in seed germination and seedlings in salinized (abandoned farmland and meadow) and non-salinized (corn farmland edge) soil habitats under different sodium chloride (NaCl) concentrations. Under NaCl stress, compared with *G. uralensis* in non-salinized habitats, seeds from salinized habitats germinated earlier, with higher germination rates and germination speeds. Under NaCl stress, seedlings from salinized habitats exhibited higher fresh weights and stronger water-holding capacities, as well as less accumulation of toxic malondialdehyde and less leakage of cell contents. The active oxygen contents of seedlings from salinized habitats were lower, while their antioxidant enzyme activities were higher. Comprehensive evaluation of each index showed that the salt tolerance of *G. uralensis* seeds from salinized habitats was superior. Our findings revealed significant differences between the salt tolerance capacities of *G. uralensis* germplasms from different habitats and indicated that, compared to non-salinized populations, *G. uralensis* seeds from salinized habitats showed better germination and seedling growth under NaCl stress. We provide a scientific basis for screening salt-tolerant licorice germplasms.

**Abstract:**

(1) Wild *Glycyrrhiza uralensis* Fisch (licorice) seeds from different habitats are often mixed for cultivation. However, differences in the responses of seeds from different habitats to salt at the early-stage offspring stage are unclear. (2) Our objective was to evaluate the salt tolerance of *G. uralensis* germplasms by comparing differences in seed germination and seedling vigor in salinized (abandoned farmland and meadow) and non-salinized (corn farmland edge) soil habitats under different sodium chloride (NaCl) concentrations. (3) The germination rates and germination indexes of seeds from the two salinized habitats with 0–320 mmol·L^−1^ NaCl were higher and their germination initiation times were earlier. Only seeds from salinized habitats were able to elongate their germs at 240 mmol·L^−1^ NaCl. Seedlings from salinized habitats had higher fresh weights and relative water contents, while they exhibited lower accumulation of malondialdehyde and less cell electrolyte leakages. Under NaCl treatment, seedlings from the salinized habitats displayed higher superoxide dismutase, catalase, and peroxidase (SOD, CAT, and POD) activities and lower superoxide anion and hydrogen peroxide (O_2_^−^ and H_2_O_2_) contents. Their comprehensive scores showed that the vigor of licorice seeds from salinized habitats was higher. (4) The salt tolerances of different wild *G. uralensis* seeds were different, and the offspring of licorice from salinized habitats had stronger early-stage salt tolerances.

## 1. Introduction

*Glycyrrhiza uralensis* Fisch (licorice) is a perennial herb belonging to the Fabaceae (Leguminosae) family [1]. Its roots contain glycyrrhizic acid, glycyrrhetinic acid, and a variety of flavonoids, which exert anti-inflammatory effects [2], soothe cough [3], and inhibit cancer cell proliferation [4]. The sweetness of glycyrrhizic acid and glycyrrhetinic acid, a natural sweetener that is widely used in the food industry, is 200 times that of sucrose [5]. Glabridin is favored by the cosmetics industry due to its antioxidant and skin-whitening effects [6]. Because of its strong salt tolerance, it can be used to improve salinized soil, thus exhibiting obvious ecological value. Long-term excavations by predators have led to the extinction of wild licorice in China. Therefore, cultivated licorice is gradually replacing wild-type licorice.

Planting licorice in salinized–alkali deserts or salinized abandoned farmland is considered to be advantageous both ecologically and economically because it can effectively resolve the competition on limited farmland between licorice and other crops [7]. Although mature *G. uralensis* plants exhibit strong salt tolerances and often act as a constructive species in halophytic meadows [8,9], they show relatively low salt tolerances during their early stages (i.e., seed germination and seedling colonization stages) [10,11]. The seed emergence rates and seedling survival rates of *G. uralensis* planted in salinized deserts or abandoned farmland are relatively low, thereby seriously affecting the sustainability of the *G. uralensis* industry. Due to natural selection and maternal effects, plants inhabiting salinized soil habitats for long periods tend to exhibit stronger salt tolerance than plants inhabiting non-salinized environments [12,13]. The habitat of wild *G. uralensis* is broad, including both salinized and non-salinized habitats. At present, the seeds of wild *G. uralensis* from different habitats are often mixed together after being harvested as a source of seeds for cultivated *G. uralensis*. However, whether the maternal effects of salinized soil cause a significant difference between the salt tolerances of licorice seeds from salinized soil habitats and those from non-salinized soils remains unreported.

The germination period, which represents the initial stage in the life history of seed plants, is also a crucial phase [14,15]. The germination rate, germination index, and growth of seedlings determine whether a species is able to effectively colonize a habitat. Osmotic-pressure-related effects caused by salt stress may inhibit licorice seed imbibition, delay their germination, and reduce their germination rate and germination index, resulting in an uneven emergence and a dearth of seedlings in the field [16]. Under normal circumstances, reactive oxygen species (ROS), such as the superoxide anion (O_2_^−^) and hydrogen peroxide (H_2_O_2_), are in a dynamic balance, but after plants are subjected to salt stress, they will accumulate in large quantities. This will affect the normal metabolism of plants through the peroxidation of nucleic acids, proteins, and oils. At the same time, reactive oxygen species also destroy the selectivity of biofilms, resulting in increased membrane permeability and membrane lipid peroxidation, and ultimately causing different degrees of damage or even death to plants [17,18]. Plant salt tolerance is a multi-pathway induction process. For example, during their long-term evolutionary process, plants evolved complex antioxidant systems, including enzymes such as superoxide dismutase (SOD), catalase (CAT), and peroxidase (POD) to scavenge ROS [19,20]. Therefore, this study compared the differences in the physiological parameters associated with seed germination and seedling growth of *G. uralensis* from different habitats in order to evaluate the salt tolerances of different germplasms. In addition, the activities of antioxidant enzymes and the ROS contents of seedlings from different populations were determined to compare the antioxidant capacities of the three germplasms and provide a scientific basis for screening salt-tolerant licorice germplasms.

## 2. Materials and Methods

### 2.1. Materials

*G. uralensis* seeds were collected from Buerjin County, Xinjiang (N 47° 54.9234′ E 86° 50.2838′) in October 2021. The non-salinized habitat was located at the edge of a corn field (salt concentration of the soil was 0.4 g/kg), while the two salinized habitats were located on salinized abandoned farmland (salt concentration of the soil was 4.2 g/kg) adjacent to the corn field and a salinized meadow (soil salt concentration was as high as 10.1 g/kg) 2.2 km away from it. Healthy mature seeds were randomly collected from these three populations, placed in paper bags, and stored in a refrigerator at −4 °C.

### 2.2. Experimental Design

Healthy and plump seeds of *G. uralensis* collected from the above three populations were selected, soaked in 98% sulfuric acid solution for 30 min, and then rinsed with running water until there was no sulfuric acid residue remained on the surface. The surfaces of the seeds were disinfected with 0.3% potassium permanganate solution for 10 min, and then rinsed with running water again until no potassium permanganate residue remained on the surfaces, following which water on the surfaces of the seeds was dried with filter paper. A total of 30 seeds were evenly placed in a 9 cm diameter glass dish with a double-layer filter paper. Five milliliters of sodium chloride (NaCl) solution of 0 (distilled water), 80, 160, 240, and 320 mmol·L^−1^ was added to Petri dishes, and germination tests were conducted in the dark in 15 °C/12 h and 25 °C/12 h artificial climate chambers (PQX-330A-12H, Ningbo Laifu Technology Co., Ltd., Ningbo, China). Each of the 15 treatments was replicated thrice. During the test, filter paper was replaced every two days, and the NaCl solution was replaced to prevent the growth of mold. Water lost due to evaporation was replenished with sterile water to maintain a constant salt solution concentration via daily weighing. The number of germinated seeds was counted every day for 7 days. After the germination test was completed, the ungerminated seeds were selected, rinsed with distilled water until there was no residual NaCl solution on the surface, and evenly placed in another Petri dish containing a double-layer filter paper. Five milliliters of distilled water was added to each Petri dish, and the seed germination test was continued for 7 days under the above-stated temperature and light conditions.

### 2.3. Analysis of Salt Concentration in Original Habitat

Collection of *G. uralensis* seeds from these three populations involved collecting a number of samples from the soil surface (0–10 cm) using the five-point sampling method. Following Chen’s method [21], a conductivity meter (DDS-11A, Shanghai Leici Laboratory Desktop Digital Conductivity Meter, Shanghai, China) was used to measure the electrical conductivity (EC) of the soil solutions. Total water-soluble salt contents of the soil samples were calculated, and a pH meter was used to determine soil pH value.

### 2.4. Calculation of Seed Germination Parameters

The seed germination rate, germination index, and germination recovery rate were calculated using a criterion based on the radicle penetrating the seed coat by at least 1 mm.

The germination rate (Gr) was calculated as the ratio of the number of seeds that germinated within 7 d to the total number of seeds.
Germination index (Gi) = ∑ Gt/D.
Germination recovery rate = (a − b)/(c − b) × 100%.
where a is the number of seeds that germinated after being transferred to distilled water, b is the number of seeds that germinated in different concentrations of NaCl solution, and c is the total number of seeds.

Final germination percentage (FGp): the total number of seeds germinated in 14 days.

### 2.5. Determination of Seed Embryo and Seed Coat Biomass

Thirty healthy and full *G. uralensis* seeds from the three populations were selected and transferred to Petri dishes with water. Seed coats and embryos were carefully separated using tweezers, dried to a constant weight in an oven at 70 °C, and weighed using an analytical balance. Each population was measured thrice.

### 2.6. Determination of Seedling-Related Parameters

At the end of the germination experiment, we randomly selected 10 seedlings from each treatment and measured their fresh weight (FW) (g), as well as their germ length (Gl) (cm) and radicle length (Rl) (cm) with a vernier caliper, measured their relative electrical conductivity (REC) (%) with an electrical conductivity meter, measured and calculated their malondialdehyde (MDA) content using the thiobarbituric acid method [22], measured their superoxide dismutase (SOD), catalase (CAT), and peroxidase (POD) activity, and evaluated their superoxide anion (O_2_^−^) and hydrogen peroxide (H_2_O_2_) content using a (Solarbio) enzyme activity kit [23]. Relative water content (RWC) was calculated as relative water content (RWC) (%) = fresh weight (g) − dry weight (g)/turgid weight (g) − dry weigh (g) × 100% [24]. The indexes for each treatment group were measured thrice.

### 2.7. Data Analysis

All statistical analyses were performed using SPSS 25.0. One-way ANOVA followed by Duncan’s multiple comparison method were used to test differences between treatments at a 5% probability level. A histogram was drawn using Origin Pro 2017 with the average value of the column height characterization parameters and the standard error of the average value of the rod characterization. Sangerbox http://sangerbox.com/home.html. (accessed on 2 January 2024) was used to analyze the correlation of various indicators and create correlation graphs. SPSS 20.0 software was used to standardize the original data (Z-score method). Then, the applicability of factor analysis was carried out by a Kaiser–Meyer–Olkin (KMO) test and Bartlett sphere test. The principal component was solved from the correlation matrix, and then the principal component coefficient, principal component score, and comprehensive score were calculated.

## 3. Results

### 3.1. Differences in Seed Germination Ability of Different Populations

Evaluation of the total salt content of soil in the original habitats of the three *G. uralensis* populations indicated that the total soluble salt contents of the salinized abandoned farmland and salinized meadow soil were 4.2 and 10.1 g per kg, respectively, whereas the total soluble salt content of the soil from the edge of the corn farmland was only 0.4 g per kg (Table 1).

Germination rate directly embodies seed vigor. The seed germination rates of the three populations gradually decreased with increasing NaCl concentration. The germination rate of seeds from the salinized abandoned farmland and edge of corn farmland decreased significantly with 240 mmol·L^−1^ NaCl treatment, while the germination rate of the seeds from the salinized meadow did not decrease significantly until the NaCl treatment reached 320 mmol·L^−1^. There was no significant difference between the germination rates of seeds from salinized abandoned farmland and salinized meadow under NaCl concentrations other than those corresponding to the 240 mmol·L^−1^ NaCl treatment. At similar NaCl concentrations, the germination rates of seeds from salinized habitats were significantly higher than those from non-salinized habitats (Figure 1A).

The germination index, which represents the germination speed of seeds, is an important indicator of seed vigor. The germination index of the three populations decreased gradually with increasing NaCl concentrations, and the germination indexes of seeds from salinized abandoned farmland and the salinized meadow were similar under the same NaCl concentrations, while they were significantly higher than those of seeds from non-salinized habitats (Figure 1B).

The cumulative germination rate reflects daily germination of seeds over time. The seeds of *G. uralensis* from salinized abandoned farmland germinated on the first day following 80 mmol·L^−1^ NaCl treatment, and there was no significant difference between the germination rate on that day and the final germination rate (Figure 2B). The germination rate of seeds from the three populations was gradually delayed with increasing NaCl concentration, with the delay in the germination of seeds from non-salinized habitats being the most significant. The seeds from non-salinized habitats did not germinate on the first day following treatment with each concentration of NaCl, its delay effect being most obvious with 320 mmol·L^−1^ NaCl treatment. A small number of seeds germinated on the third day, but the germination rate was only 20% (Figure 2A).

The germination test indicated that the germination rate of seeds from salinized habitats (salinized abandoned farmland and salinized meadow) reached 70–96%, with the germination recovery test indicating that almost all seeds that did not germinate recovered later. The final germination rate of both reached almost 100% on day 14 (Figure 3B,C). However, the germination test showed that the germination rate of seeds from non-salinized habitats was only 39–70% and that only approximately 30–60% of the ungerminated seeds recovered sufficiently to germinate. The final germination percentage on day 14 was only 83–90% (Figure 3A).

### 3.2. Differences in Seed Coats from Different Habitats

The germination test showed that the germination rates and germination indexes of seeds from different populations were significantly different. Therefore, we dissected the imbibed seeds of the three populations. The results showed that there was no significant difference among the embryo dry weights of the three populations; however, there were significant differences in seed coat quality among the different populations. The seed coat quality of seeds from salinized abandoned farmland was the highest, while that of seeds from non-salinized habitats was the lowest (Figure 4).

### 3.3. Differences in the Development and Physiological Indexes of Seedlings from Different Populations

The fresh weight of seedlings reflects their vitality, while relative water content reflects the ability of plants to retain water under salt stress. Analysis of variance (ANOVA) indicated significant differences among the fresh weights of seedlings from different NaCl treatment groups. In the control group, the fresh weight biomass of seedlings from the salinized meadow was the largest (0.1 g). The fresh weights of seedlings from the three populations decreased gradually with increasing NaCl treatment concentration. The fresh weights and relative water contents of the seedlings from the salinized meadow were comparable to those from the salinized abandoned farmland and also significantly higher than those of the seeds from the non-salinized habitat. The fresh weight of seedlings from the salinized meadow was 1.3–1.5 times higher than that of seedlings from the non-salinized habitat (Figure 5A,B).

Relative electrolyte leakage reflects cell membrane permeability. The relative electrolyte leakage of *G. uralensis* seedlings differed significantly among treatments. The relative electrolyte leakage of the seedlings from the three populations increased gradually with increasing NaCl concentration, indicating that the membrane permeability of the seedlings from the three populations had increased significantly due to increasing NaCl concentration in the environment. At a NaCl concentration range of 160–320 mmol·L^−1^, the relative electrolyte leakage of seedlings from the non-salinized habitat was the highest, while that from salinized meadows was the lowest, indicating that under moderate or high salt stress, the leakage of electrolytes from seedlings in non-salinized habitats would be the most serious, while the seedlings from salinized meadows were the least damaged (Figure 5C).

The content of MDA reflects the degree of membrane lipid peroxidation. MDA content in *G. uralensis* seedlings was significantly different among treatments. At a low concentration of NaCl treatment (80–160 mmol·L^−1^), MDA contents in the seedlings of all populations were lower, relatively, and the differences were non-significant. The MDA content of the seedlings from the three populations increased gradually with increasing NaCl concentration. A large amount of MDA accumulated under high NaCl concentration (240–320 mmol·L^−1^) stress. The MDA content of seedlings from non-salinized habitats was the highest, whereas that of seedlings from salinized meadows was the lowest (Figure 5D).

Maximum germ and radicle lengths of seedlings from non-salinized habitats were observed in the control group. The germ and radicle lengths of the seedlings from the three populations decreased significantly with increasing NaCl concentration. At a salt concentration of 240 mmol·L^−1^, the seeds from the salinized meadow and the salinized abandoned farmland developed into intact seedlings with radicles and germs. However, seeds from non-salinized habitats could not form germs that broke through the seed coat (Figure 5E).

### 3.4. Differences in the Antioxidant Capacities of Seedlings from Different Populations

O_2_^−^ and H_2_O_2_ contents and SOD and POD activities of *G. uralensis* seedlings in the three populations differed with increasing NaCl concentration, indicating that the antioxidant enzyme activities of seedlings of *G. uralensis* may change according to the habitat of the mother. The O_2_^−^ content in the seedlings of different populations showed a gradual upward trend with increasing NaCl concentration, while the H_2_O_2_ content showed a trend of first decreasing and then increasing. Under similar NaCl concentrations, the contents of O_2_^−^ and H_2_O_2_ in the seedlings from salinized abandoned farmland and salinized meadows were lower, whereas the activities of SOD, CAT, and POD in these seedlings were higher compared to those from non-salinized habitats (Figure 6A–E).

### 3.5. Comprehensive Evaluation of Germplasms from Different Populations

In order to further explore the relationship among various indicators in the early stages of licorice life history and to analyze the main contribution sources of salt tolerance of seeds, this experiment involved Pearson correlation analysis on 14 indexes (Gi, REC, Fw, MDA, O_2_^−^, Rl, Gl, RWC, Gr, FGr, CAT, H_2_O_2_, POD, and SOD) during the germination stage of seeds.

The KMO value obtained through KMO testing is 0.843, indicating that there is a certain correlation between the indicators. The Bartlett sphere test result is 454.142, and the Sig value is 0.000, which indicates that it can be used for subsequent comprehensive score analysis. The results showed that there was a highly significant positive correlation among Gi, FGr, Fw, RWC, and Gr, which have a significant negative correlation with the REC, MDA, and O_2_^−^ (Figure 7).

Principal component analysis was performed on the above 14 indexes. The first three principal components explained 77.585% of the total variance, indicating that the extracted three principal components can represent 77.585% of the original 14 indexes. The extracted principal components have certain significance for evaluating different treatments. Therefore, three principal components were extracted, which were Y1, Y2, and Y3, respectively. Y4 is the composite score (Table 2).
Y1 = 0.34211*X*_1_ − 0.34023*X*_2_ + 0.33874*X*_3_ − 0.32824*X*_4_ − 0.32562*X*_5_ + 0.30614*X*_6_ + 0.30201*X*_7_ + 0.28703*X*_8_ + 0.25068*X*_9_ + 0.18473*X*_10_ − 0.16937*X*_11_ − 0.00786*X*_12_ + 0.03152*X*_13_ − 0.16787*X*_14_.
Y2 = 0.12002*X*_1_ + 0.15657*X*_2_ − 0.12977*X*_3_ − 0.02437*X*_4_ − 0.16998*X*_5_ − 0.24248*X*_6_ + 0.29366*X*_7_ − 0.03289*X*_8_ − 0.37591*X*_9_ + 0.44902*X*_10_ − 0.41795*X*_11_ − 0.35641*X*_12_ + 0.31864*X*_13_ − 0.14987*X*_14_.
Y3 = 0.00684*X*_1_ − 0.04204*X*_2_ − 0.00586*X*_3_ + 0.04008*X*_4_ − 0.11244*X*_5_ − 0.11537*X*_6_ + 0.18870*X*_7_ + 0.38817*X*_8_ − 0.09679*X*_9_ + 0.2679*X*_10_ + 0.26497*X*_11_ + 0.38328*X*_12_ − 0.41163*X*_13_ + 0.56221*X*_14_.
Y4 = 0.5087Y1 + 0.1924Y2 + 0.0746Y3.

It can be seen from the above formula that in the principal component Y1, the absolute values of the coefficients of Gi, REC, Fw, MDA, O_2_^−^, Rl, Gl, RWC, Gr, FGr, CAT, and SOD are much greater than the absolute values of the coefficients of other variables. Therefore, the principal component Y1 is a comprehensive reflection of the 12 indexes, which represents the germination vigor of *G. uralensis* seeds under salt stress. It also shows that these 12 indexes are essential elements for characterizing the salt tolerance of *G. uralensis* germplasm.

Table 3 shows the principal component score calculated according to the principal component equation and the comprehensive score calculated by the ratio of the variance contribution rate of each principal component to the total variance contribution rate of the three principal components.

The comprehensive scores of seeds from salinized habitats in the 0–160 mmol·L^−1^ NaCl treatment were near or above 1, indicating that the activity of seeds from salinized habitats was the strongest among all treatments in the range of 0–160 mmol·L^−1^ NaCl concentration. The comprehensive scores of seeds from non-salinized habitats were less than 0, which indicated that the seed vigor of *G. uralensis* from non-salinized habitats was worse than that from salinized habitats, especially under salt treatment (Table 3).

## 4. Discussion

Seed germination represents the initial stage in the life cycle of seed plants [25]. Climate change, improper fertilization, and irrigation tend to enhance soil salinity, posing new challenges to the seed germination and seedling development capacities of many land plants [26], including licorice. The inhibitory effects exerted by salt stress on seed germination manifests mainly as delays in initial seed germination, slowness of germination speed, and a decrease in germination rates [27]. Under the effect of stress exerted by 240–320 mmol·L^−1^ NaCl, *G. uralensis* seeds, derived from salinized habitats (salinized abandoned farmland and salinized meadow), exhibited higher germination rates, faster germination speeds, and earlier germination than those derived from the non-salinized habitats, indicating that the seeds of salinized habitats are able to show a higher degree of tolerance to salt stress. The high degree of similarity between the germination rates and germination processes of the two types of licorice seeds from salinized habitats indicated that seeds from salinized habitats were more adaptable to salt stress on account of the maternal effect of salinized habitats. The seedlings of *Zygophyllum coccineum* [28] and *Hippophae rhamnoides* L. [29] derived from a salinized population were more tolerant to salt stress than those from a non-salinized population, which was consistent with our results. This may account for plants growing in salinized habitats yielding offspring seeds that exhibit higher salt tolerance. However, the germination rates of *Suaeda vermiculata* [30] from salinized habitats that grew under high salt stress were lower than those from non-salinized habitats, indicating that the maternal effect may vary among plant species. Anatomical studies have shown that the seed coat biomass of licorice seeds from salinized populations was higher than that of seeds from non-salinized populations. Previous studies have shown that the coats of some seeds protect the germination of those seeds in high-salt solutions [31]. On the one hand, the coat may separate harmful ions such as sodium (Na^+^) and block their entry; on the other hand, it may prevent potassium (K^+^) leakage [32]. Wei compared the anatomical structure of wild soybean (*Glycine max*) seeds that survived in salinized and non-salinized environments and found that soybean seeds from the salinized population had thicker seed coats, larger biomass, more developed phloem fibers, and more secondary xylem [33], and their result was consistent with ours. Therefore, we believe that the higher seed coat biomass of licorice seeds from salinized populations may be an adaptation strategy aimed at salinized environments.

At the germination stage, the germination rate of seeds from non-salinized habitats only approximated 54–77%, and a considerable number of seeds did not imbibe at the rehydration germination stage. By contrast, the germination rate of seeds from salinized habitats (salinized abandoned farmland and salinized meadow) was as high as 70–96%, and almost all seeds in the NaCl treatment group that had not germinated already did so at the rehydration stage, with the final germination percentage approaching 100% on day 14. This indicated that the vitality (representing salt tolerance) of seeds from salinized habitats was stronger.

Early seedling morphogenesis, a key event in plant life history, is crucial for the subsequent growth and development of plants [34]. Normal cell membranes allow selective penetration of substances. Osmotic stress and ion toxicity caused by salt stress result in a large amount of ROS accumulating in the cell membrane, leading to changes in membrane permeability and increased membrane lipid peroxidation, as well as physiological and biochemical reactions in the seedlings [35]. The results of our study indicated that the fresh weights and relative water contents of the seedlings from the three populations decreased gradually with increasing NaCl concentration, whereas the MDA content and relative electrolyte leakage increased significantly. At similar NaCl concentrations, the relative water contents of seedlings from non-salinized habitats were significantly lower than those from salinized habitats, whereas their MDA contents and relative electrolyte leakage levels were significantly higher than those from salinized habitats. Thus, changes in the cell membrane permeability of *G. uralensis* seedlings from non-salinized habitats under salt stress were more severe, the accumulation of harmful substances was greater, water loss was more serious, and seedling vigor decreased faster; thus, germs did not form under high NaCl concentrations. Under different NaCl treatments, the fresh weight of seedlings from non-salinized habitats was lower than that of seedlings from salinized habitats.

Under salt stress, plants produce excessive ROS, which may cause metabolic disorders. In plants, SOD catalyzes the reaction 2O_2_^−^ + 2H^+^ → H_2_O_2_ + O_2_, and the H_2_O_2_ produced is dissociated by enzymes, such as POD and CAT, which protect the plant from the effects exerted by ROS. In the present study, SOD, CAT, and POD activities in the seedlings from the two salinized populations were significantly higher than those from the non-salinized populations. *Zygophyllum xanthoxylum* from salinized habitats shows higher SOD activity than that from non-salinized habitats [36]. Studies which investigated the antioxidant capacities of four ecotypes of *Phragmites australis* [37], namely, aquatic reeds, dune reeds, mildly salinized meadow reeds, and severely salinized meadow reeds, reported that SOD activity of reed seedlings from the severely salinized meadow was the highest, which is consistent with our results. Under NaCl stress, licorice seedlings from salinized habitats exhibited a higher antioxidant capacity, which helps them cope with the damage caused by ROS in long-term salt stress environments. We believe that this may reflect an adaptation to salinized environments.

The correlation analysis showed that there was a highly significant positive correlation among Gi, FGp, Fw, RWC, and Gr, so that could be used as an important index to evaluate the salt tolerance of *G. uralensis* seeds. There is also a significant negative correlation with REC, MDA, and O_2_^−^ which could characterize the stress degree of seeds under salt stress. The higher the comprehensive score, especially in the treatment greater than 1, is mostly from the salinized habitats, but the ranking of seeds from non-salinized habitats is the opposite. Therefore, the comprehensive evaluation of each index showed that the salt tolerance of *G. uralensis* seeds from salinized habitats was superior.

## 5. Conclusions

The results of the present study revealed significant differences among the salt tolerance capacities of *G. uralensis* germplasms from different habitats. Compared to non-salinized habitats, *G. uralensis* seeds from salinized habitats showed better germination ability and seedling growth under NaCl stress. *G. uralensis* seedlings from salinized habitats showed higher antioxidant enzyme activities, as well as lower ROS and MDA contents, in addition to well-preserved membrane functions. Our findings indicated that seeds from salinized habitats show a greater degree of tolerance to salt stress during germination and seedling growth. Therefore, on account of the different levels of salt tolerances shown by licorice seeds from different habitats, we suggest that these seeds should be stored separately and that licorice seeds from salinized habitats should be preferably sown on salinized land.

## Figures and Tables

**Figure 1 biology-13-00052-f001:**
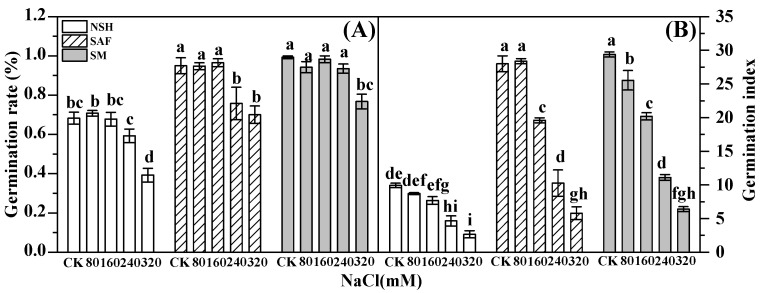
Effects of NaCl treatment on seed germination rate and germination index of three populations of *G. uralensis.* Different letters represent significant differences (*p* < 0.05) among different treatments. Germination rate (**A**), germination index (**B**).

**Figure 2 biology-13-00052-f002:**
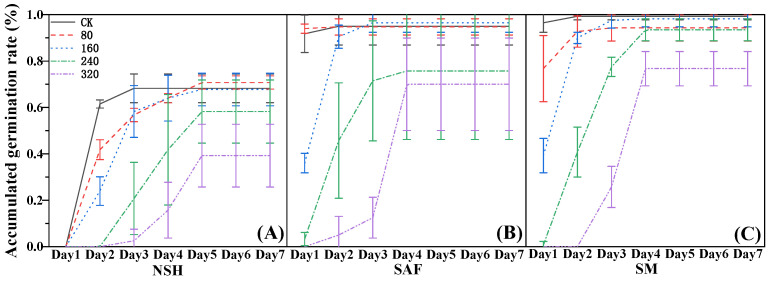
Effects of NaCl treatment on the cumulative germination rate of three populations of *G. uralensis.* Non-salinized habitats (**A**), salinized abandoned farmland (**B**), salinized meadow (**C**).

**Figure 3 biology-13-00052-f003:**
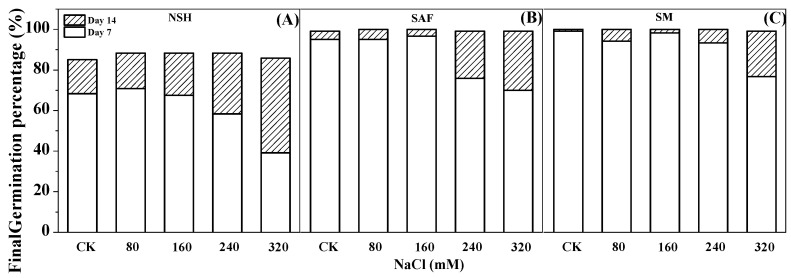
Effects of NaCl treatment on the final germination percentage (day 14) and germination rate (day 7) of seeds from three populations. Non-salinized habitats (**A**), salinized abandoned farmland (**B**), salinized meadow (**C**).

**Figure 4 biology-13-00052-f004:**
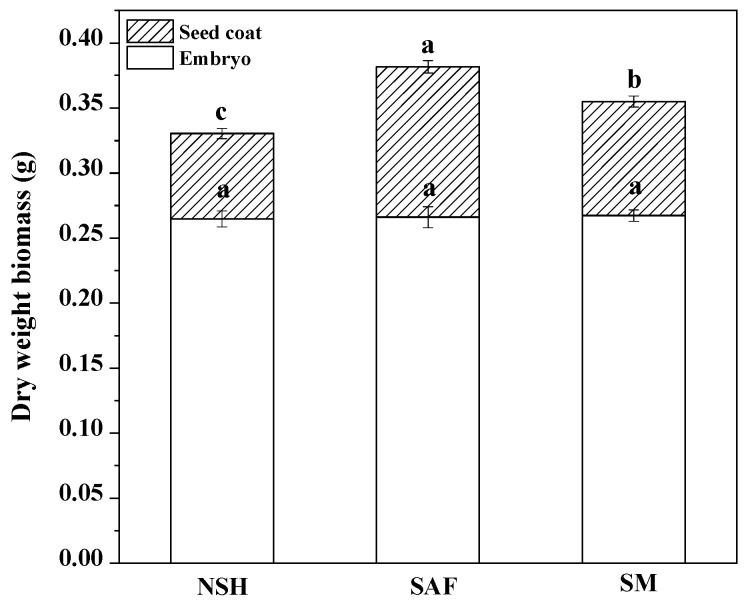
Dry weight biomass of seed coat and embryo from three populations of *G. uralensis.* Different letters represent significant differences (*p* < 0.05) among different treatments.

**Figure 5 biology-13-00052-f005:**
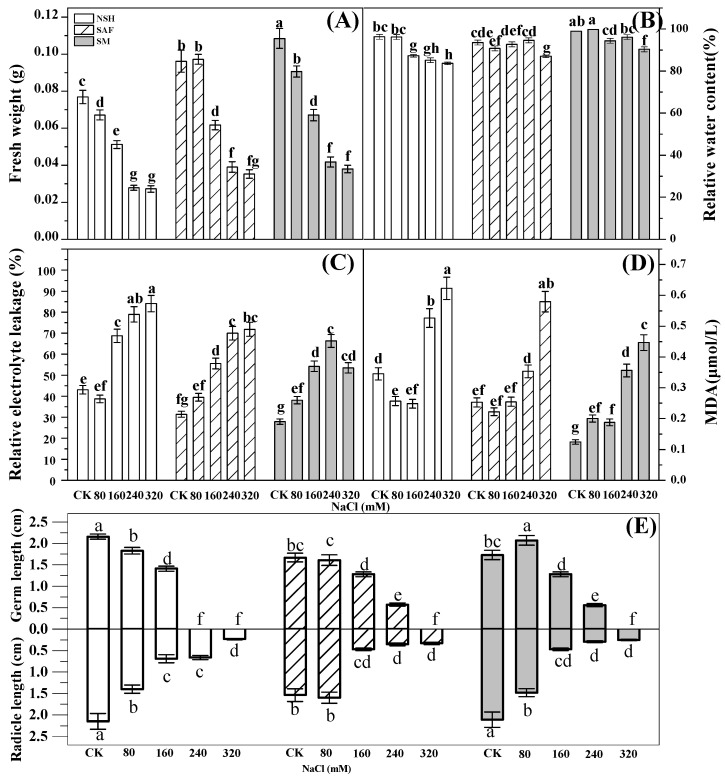
Effects of NaCl treatment on the fresh weight, relative water content, relative electrolyte leakage, MDA content, germ length, and radicle length of seedlings from three populations of *G. uralensis*. Different letters represent significant differences (*p* < 0.05) among different treatments. Fresh weight (**A**), relative water content (**B**), relative electrolyte leakage (**C**), MDA (**D**), germ length and radicle length (**E**).

**Figure 6 biology-13-00052-f006:**
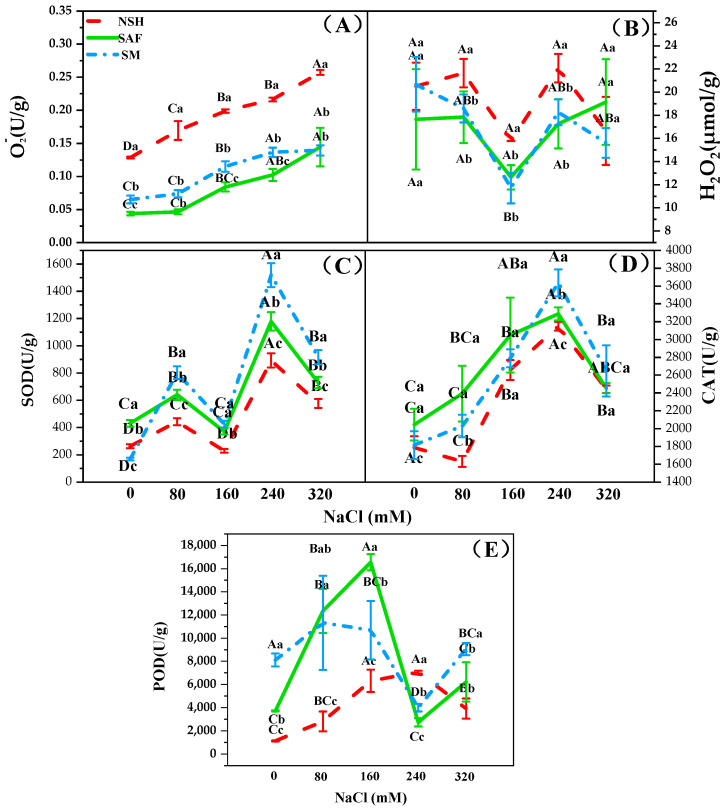
SOD, CAT, and POD activity or O_2_^−^ and H_2_O_2_ content of three populations of *G. uralensis* seedlings under NaCl treatment. Different capital letters represent significant differences (*p* < 0.05) among different NaCl concentrations in the same population. Different lowercase letters represent significant differences (*p* < 0.05) among different populations at the same NaCl concentrations. Superoxide anion (O_2_^−^, (**A**)), hydrogen peroxide (H_2_O_2_, (**B**)), superoxide dismutase (SOD, (**C**)), catalase (CAT, (**D**)), peroxidase (POD, (**E**)).

**Figure 7 biology-13-00052-f007:**
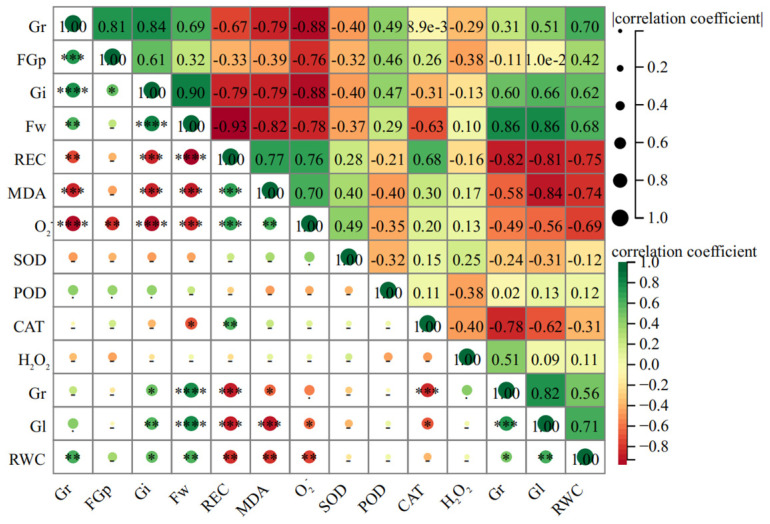
Pearson correlation analysis among 14 indexes of *G. uralensis.* “-”: no significant difference; ✽: *p* < 0.05; ✽✽: *p* < 0.01; ✽✽✽: *p* < 0.001; ✽✽✽✽: *p* < 0.0001.

**Table 1 biology-13-00052-t001:** Total water-soluble salt contents and pH of the soil in the *G. uralensis* plot.

Habitats	Salts Contents (g/kg)	pH
Non-salinized habitat (NSH)	0.4 ± 0.03	7.3 ± 0.25
Salinized abandoned farmland (SAF)	4.2 ± 0.46	8.3 ± 0.02
Salinized meadow (SM)	10.1 ± 1.2	8.4 ± 0.01

**Table 2 biology-13-00052-t002:** Contribution rate of principal component variance factor.

Explanation of Total Variance						
Formulation	Initial Eigenvalue			Extract Square and Load		
	Total	Variance %	Accumulate %	Total	Variance %	Accumulate %
1	7.122	50.872	50.872	7.122	50.872	50.872
2	2.694	19.244	70.116	2.694	19.244	70.116
3	1.046	7.469	77.585	1.046	7.469	77.585
4	0.783	5.592	83.177			
5	0.674	4.818	87.995			
6	0.523	3.734	91.729			
7	0.329	2.353	94.082			
8	0.286	2.043	96.125			
9	0.152	1.083	97.208			
10	0.131	0.936	98.144			
11	0.11	0.788	98.932			
12	0.073	0.521	99.453			
13	0.049	0.352	99.806			
14	0.027	0.194	100			

**Table 3 biology-13-00052-t003:** Principal component score and comprehensive score.

Treatment	Principal Component Y Score			Ranking Lists
	Y1	Y2	Y3	
SM Ck	3.7125	3.0275	3.055	1
SM 160	2.7525	2.245	2.2625	2
SAF Ck	2.7175	2.1225	2.2225	3
SAF 160	2.475	1.8725	2.0175	4
SM 240	1.4475	1.1675	1.1875	5
SAF 240	1.1325	0.8875	0.9275	6
SM 320	−0.0075	0.0975	0.0075	7
NSH 160	−0.1525	0.0775	−0.1	8
NSH Ck	−0.31	−0.115	−0.2375	9
SAF 320	−0.43	−0.255	−0.3375	10
SM 80	−0.9975	−0.6675	−0.8025	11
NSH 240	−1.8425	−1.5175	−1.5175	12
SAF80	−2.125	−1.73	−1.7475	13
NSH 320	−3.5875	−3.08	−2.97	14
NSH 80	−4.785	−4.14	−3.9675	15

Note: Non-salinized habitat (NSH); salinized abandoned farmland (SAF); salinized meadow (SM).

## Data Availability

The raw data supporting the conclusions of this article will be made available by the authors on request.

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
