# Peer review of "Maternal Effects of Habitats Induce Stronger Salt Tolerance in Early-Stage Offspring of *Glycyrrhiza uralensis* from Salinized Habitats Compared with Those from Non-Salinized Habitats"

_biology, 2024, doi:10.3390/biology13010052_

Round 1

Reviewer 1 Report

Comments and Suggestions for Authors

In this paper the authors evaluate the salt tolerance of salt tolerance of Glycyrrhiza uralensis by comparing differences in seed germination and seedlings vigor in salinized and non-salinized habitats. The research is valuable because in the world are many problems because of the salinized soils. However after I read the manuscript I have some issues.

At the Introduction there are present information concerning Glycyrrhiza uralensis and tolerance at salinized soils but I don't see information about similar researches. There were the studies with Glycyrrhiza uralensis tolerance at salinized soils ? Or this is the first study ?

The Methodology are well described, however I have a little issue, how many non-salinized habitats are in this study ? 

The results are well presented, I don't have any issue. 

At Discussions I have same question like at Introduction.

I am agree with publishing of this paper with minor modifications.

Comments on the Quality of English Language

English language are fine

Author Response

Dear reviewers, according to your comments on the article has been modified, the quality of the article has been greatly improved, very grateful for your help to us, the following is a response to your questions.

  1. At the Introduction there are present information concerning Glycyrrhiza uralensis and tolerance at salinized soils but I don't see information about similar researches. There were the studies with Glycyrrhiza uralensistolerance at salinized soils ? Or this is the first study ?

The salt tolerance of Glycyrrhiza uralensis has been widely studied, and I have added articles on the salt tolerance of Glycyrrhiza uralensis to the background.

  1. The Methodology are well described, however I have a little issue, how many non-salinized habitats are in this study ?

This study included only one non-salinized habitat from maize farmland.

  1. At Discussions I have same question like at Introduction.

In the discussion, I have supplemented the relevant references on the tolerance of licorice seeds and seedlings to salt stress.

Reviewer 2 Report

Comments and Suggestions for Authors

Dear author,

I reviewed your manuscript and made some observations and recommendations. I hope that these will be useful for improving the manuscript. Attached is the revised manuscript.

Thank you for colaboration,

Happy New Year!

Author Response

Dear reviewers, according to your comments on the article has been modified, the quality of the article has been greatly improved, very grateful for your help to us, the following is a response to your questions.

Reviewer 3 Report

Comments and Suggestions for Authors

In this article, the authors conducted a comparison of physiological parameters related to seed germination and seedling growth in G. uralensis from varied habitats to assess the salt tolerance among different germplasms. Additionally, this research included the determination of antioxidant enzyme activity and ROS content in seedlings across diverse populations. I think this study included some interesting data. Below are a series of comments and suggestions, in no particular order.

1.    Licorice is a plant that undergoes self-pollination. The authors should demonstrate the consistency of the genetic background of the Licorice they used.

2.    The authors should show the kind and the content of different inorganic salts in the soil, for example NaCl, Na2CO3, CaCl2, MgSO4, Na2SO4, I think they may influence the results.

3.    I strongly suggest the author to put the data of OH- and total ROS level.

4.    The chemical formula for the superoxide ion was written incorrectly. (From line 275). The authors should check their article carefully.

5.    The authors only collected seeds from single year and single place. I think it lacks representativeness.

6.    The authors should present some pictures, especially pictures of the seedlings, to illustrate their conclusions.

7.    Please add some information about the relationship between salt stress and ROS in the introduction part.

8.    Line 287 The author should clarify the distinction between capital letters and lowercase letters in Figure 6.

9.    Line 297 The author should tell us what is the KOM value.

10. Line 356 It is better to cite the previous studies.

11. Line 362 It seems like there might be a grammar issue in this sentence.

Comments on the Quality of English Language

No

Author Response

Dear reviewers, according to your comments on the article has been modified, the quality of the article has been greatly improved, very grateful for your help to us, the following is a response to your questions.

  1. Licorice is a plant that undergoes self-pollination. The authors should demonstrate the consistency of the genetic background of the Licorice they used.

Although licorice is a self-pollinated plant, licorice has formed many hybrid lines in the wild through insect pollination and other ways. We chose different habitats in the same place to ensure that they use the same gene bank as much as possible to ensure that their genetic backgrounds are consistent.

  1. The authors should show the kind and the content of different inorganic salts in the soil, for example NaCl, Na2CO3, CaCl2, MgSO4, Na2SO4, I think they may influence the results.

The main purpose of this paper is to study whether the maternal effect is helpful to improve the salt tolerance of Glycyrrhiza uralensis, so as to screen the germplasm of G. uralensis with strong salt tolerance. According to previous studies, we prefer that the soil salt concentration in the maternal habitat is the primary reason for the intensity of the maternal effect. Although the parent effects of different inorganic salts in soil may be different, this is not within the scope of our research.

  1. I strongly suggest the author to put the data of OH- and total ROS level.

After consulting the literature, we selected the typical representative reactive oxygen species indicators O2-and H2O2. In addition, the changes of O2-and H2O2 content are closely related to the activities of SOD, CAT and POD. Their changes help us better understand how the reactive oxygen species of different populations of seedlings are scavenged, and better compare the differences between the three populations of seedlings in the process of scavenging reactive oxygen species.

  1. The chemical formula for the superoxide ion was written incorrectly. (From line 275). The authors should check their article carefully.

 We have made corrections

  1. The authors only collected seeds from single year and single place. I think it lacks representativeness.

The licorice seeds collected in this paper are from different populations in the same place to ensure the consistency of the background of different populations of licorice seeds as much as possible.

If licorice seeds from different locations were collected, the soil samples from different locations would be inconsistent in the type of salt, and the effects of different types of salt on the germination potential of seeds would be different. Therefore, in order to avoid other interference factors except salt concentration, we selected different populations at the same point.

Similarly, the seeds of different years were collected and mixed together, because the germination potential of licorice seeds with different storage years was different, which would also affect the test results.

The theme of this paper is : based on whether the maternal effect mediated by soil salt contributes to the improvement of salt tolerance in G. uralensis, in order to screen the germplasm of G. uralensis with strong salt tolerance. Collecting seeds from different locations does not serve my research well. For this reason, we only considered the seeds of G. uralensis collected in the same location and in the same year.

  1. The authors should present some pictures, especially pictures of the seedlings, to illustrate their conclusions.

We also wanted to show some pictures at the beginning. However, we have a total of 15 treatments, each treatment if displayed at least 10 seedlings, if all displayed, the picture will be too much, not beautiful. It is not professional enough to show the differences of seedlings of different populations under high concentration salt treatment. After consideration, we gave up the idea of showing pictures.

  1. Please add some information about the relationship between salt stress and ROS in the introduction part.

We have appropriately added the relationship between salt stress and reactive oxygen species in the background.

  1. Line 287 The author should clarify the distinction between capital letters and lowercase letters in Figure 6.

We have made changes.

  1. Line 297 The author should tell us what is the KOM value.

The KOM index refers to a measure of whether the data can be used for factor analysis. It is between 0-1, greater than 0.5 can be used for subsequent factor analysis, and the larger the data, the stronger the applicability of the data to factor analysis. In the method, we use the KOM test to obtain the KOM value. We have modified the content of 297 lines in the text.

  1. Line 356 It is better to cite the previous studies.

We have modified the references.

  1. Line 362 It seems like there might be a grammar issue in this sentence.

We have modified the grammar of the sentence.

Round 2

Reviewer 3 Report

Comments and Suggestions for Authors

I still hope that the authors could put some representative images of seedlings in this article, even if in the supplementary data.

Comments on the Quality of English Language

No

Author Response

Thank you very much indeed for your warmly suggestion, but we are very sorry that we did not take photos of the seedlings during the initial experiment. However, we guarantee that all experimental data in our manuscript is true. We hope to receive your understanding. Finally, thank you again for your valuable suggestions on our paper, we’ll pay extra attention to similar issues in the future.